# Gridless Underdetermined Direction of Arrival Estimation in Sparse Circular Array Using Inverse Beamspace Transformation

**DOI:** 10.3390/s22082864

**Published:** 2022-04-08

**Authors:** Ye Tian, Yonghui Huang, Xiaoxu Zhang, Xiaogang Tang

**Affiliations:** 1National Space Science Center, Chinese Academy of Sciences, Beijing 100190, China; tianye171@mails.ucas.ac.cn (Y.T.); zhangxiaoxu19@mails.ucas.ac.cn (X.Z.); 2University of Chinese Academy of Sciences, Beijing 100190, China; 3School of Aerospace Information, Space Engineering University, Huairou District, Beijing 101416, China

**Keywords:** DOA estimation, underdetermined, beamspace, gridless, GLS, SCA, UCA

## Abstract

Underdetermined DOA estimation, which means estimating more sources than sensors, is a challenging problem in the array signal processing community. This paper proposes a novel algorithm that extends the underdetermined DOA estimation in a Sparse Circular Array (SCA). We formulate this problem as a matrix completion problem. Meanwhile, we propose an inverse beamspace transformation combined with the Gridless SPICE (GLS) algorithm to complete the covariance matrix sampled by SCA. The DOAs are then obtained by solving a polynomial equation with using the Root-MUSIC algorithm. The proposed algorithm is named GSCA. Monte-Carlo simulations are performed to evaluate the GSCA algorithm, the spatial spectrum plots and RMSE curves demonstrated that the GSCA algorithm can give reasonable results of underdetermined DOA estimation in SCA. Meanwhile, the performance of the algorithm under various configurations of SCA is also evaluated. Numerical results indicated that the GSCA algorithm can provide access to solve the DOA estimation problem in Uniform Circular Array (UCA) when random sensor failures occur.

## 1. Introduction

Estimating the directions of arrival (DOAs) of signals with a sensor array is one of the critical research topics in the field of array signal processing, and plays an indispensable role in applications such as underwater object detection with sonar [1], radar target localization [2], wireless communications [3] and radio astronomy systems [4], and so forth.

In particular, the performance of the DOA estimation system is significantly constrained by the number of sensors. For example, classical high-resolution DOA algorithms including MUSIC [5], ESPRIT [6], and maximum likelihood (ML) type methods [7] can distinguish up to N−1 sources with *N* sensors. However, increasing the number of sensors will also increase the system complexity and cost. Furthermore, occasional sensor failures in array [8,9] severely degrade the DOA estimation performance. In this complex scenario, the number of sources may be greater than or equal to the number of sensors, resulting in a problem of underdetermined DOA estimation.

A sparse array combined with a sparse recovery algorithm offers a novel perspective on solving this intractable underdetermined DOA estimation problem [10,11]. Notably, array configurations play an important role in the DOA estimation system. Various kinds of sparse arrays have been studied intensively under the framework of the sparse recovery algorithm, such as the coprime array [12], nested array [13] and super nested array [14], and so forth. However, sensors are located along a line in the above-mentioned array configurations, which discards the scenarios of sensors being placed in a two-dimensional plane.

Among various planar arrays, the Uniform Circular Array (UCA) has attracted much attention due to its advantages of covering a $360^{\circ }$ azimuthal field of view and being easy to conform to cylindrical structures. Therefore, the UCA is widely adopted in systems such as Massive MIMO [15], Ground-Based Radar [16], and the tracking of Unmanned Aerial Vehicles (UAVs) [17], and so forth. Yet the UCA degenerates into a Sparse Circular Array (SCA) when sensor failures occur. The underdetermined DOA estimation in SCA is still an open problem. Following the idea of the nested sparse linear array (NSLA), a DOA estimation algorithm in the nested sparse circular array (NSCA) has been proposed in [18]; it is worth noting that a $L_{1}$ norm based sparse recovery method is adopted in [18,19,20]. Apparently, in this regime where a pre-defined dense grid dictionary of array steering vectors evaluated at the concerned DOA angle range is necessary, a grid mismatch problem is caused. Moreover, hyperparameters introduced in the above algorithms significantly impact the performance, and the selection of those hyperparameters is not mentioned in those papers [18,21,22].

Obviously, a gridless hyperparameter-free method for underdetermined DOA estimation in SCA is urgently needed. The idea of gridless sparse recovery, firstly introduced in [23], has received great attention from the spectral estimation community. Various prominent algorithms have been developed such as atomic norm minimization (ANM) [24,25], enhanced matrix completion (EMaC) [26], and the covariance fitting type method named gridless SPICE (GLS) [27], and so forth. The GLS has been adopted intensively among the above due to its outstanding ability to cope with multiple measurement vectors (MMV) and a hyperparameter-free property.

Nevertheless, GLS is based on finding a Toeplitz or Hankel structured matrix to fit the sample covariance matrix and interpolates the missing samples simultaneously, which cannot be satisfied in SCA. The non-Vandermonde-structured steering vector of SCA creates a big obstacle to the application of the GLS algorithm. Fortunately, a method named beam space transformation (BT) which transforms the steering vector of the UCA into a virtual Vandermonde-structured steering vector has been proposed in [28] and utilized in [20,29]. Inspired by this, we propose a gridless hyperparameter-free algorithm based on the inverse beamspace (IBT) transformation of the Toeplitz matrix which offers a way to adopt the GLS algorithm in the SCA . The sample covariance matrix of SCA in element space is completed to a Toeplitz matrix in beam space, which also provides convenience for the application of the efficient Root-MUSIC algorithm[2]. Computer simulations are performed which demonstrate the ability of the proposed algorithm to handle the tricky underdetermined DOA estimation problem in SCA. We summarize the differences and connections between our work and other related works in Table 1.

The main contributions of this work are summarized as follows:1.We propose an inverse beam space transformation (IBT) of the Toeplitz matrix in SCA scenario. The missing elements in the sample covariance of SCA are completed;2.A gridless hyperparameter-free algorithm is proposed to cope with the underdetermined DOA estimation problem in SCA. The efficient outstanding Root-MUSIC method based on the completed covariance matrix can be adopted;3.Numerical simulations are performed under various scenarios to evaluate the proposed GSCA (**G**ridless DOA Estimation in **S**parse **C**ircular **A**rray) algorithm.

***Notations***: In this paper, superscripts ${\left(\cdot \right)}^{-1}$ , ${\left(\cdot \right)}^{*}$ , ${\left(\cdot \right)}^{T}$, and ${\left(\cdot \right)}^{H}$ denote the inverse operation, complex conjugate, transpose, and conjugate transpose, respectively; ${\left(\cdot \right)}^{†}$ denotes the pseudo-inverse of a matrix. diag{·}, Toep{·}, and Tr{·} are the diagonal matrice operator, Toeplitz matrix operator, and trace operator, respectively. δ(·) is the Delta function. Boldface lowercase letters such as a, b denote vectors, and boldface uppercase letters such as A, B denote matrices, and ${\left[A\right]}_{i,j}$ denotes the (i,j)-th component of matrix A. $I_{N}$ is the N×N identity matrix. ∠z means taking the argument of the complex number *z*.

The remainder of this paper is organized as follows. Section 2.1 describes the signal model of SCA, and the GLS algorithm is introduced in Section 2.2. The inverse beamspace transformation is introduced in Section 2.3. The proposed algorithm is introduced in Section 3. The simulation results and related discussions are included in Section 4. Finally, Section 5 concludes this paper.

## 2. System Model

### 2.1. Signal Model for SCA

As shown in Figure 1, we consider an SCA that is composed of $N_{p}$ physical sensors selected from a *N*-element UCA with radius *R*. The *n*-th angle coordinate of the element located on the UCA is given by:(1)$$\alpha \left(n\right)=\frac{2\pi (n-1)}{N}.$$

Let Ω be the coordinate index set of $N_{p}$ integers selected from integers {1,2,⋯,N}, and the angle coordinates generated by Ω are represented as follows:(2)$$\beta_{i}=\alpha \left({\left[\Omega \right]}_{i}\right),$$
where ${\left[\Omega \right]}_{i}$ is the *i*-th smallest number in the set Ω. Assume *D* far-field narrowband sources with azimuthal DOAs $\phi ={\left[\phi_{1},\phi_{2},\cdots ,\phi_{D}\right]}^{T}$ impinging on the SCA. The *k*-th observed snapshot is modeled as:(3)$$y_{p}\left(k\right)=A_{p}\left(\phi \right)s\left(k\right)+n\left(k\right),\;k=1,2,\cdots ,K$$
where $s\left(k\right)={\left[s_{1}\left(k\right),s_{2}\left(k\right),\cdots ,s_{D}\left(k\right)\right]}^{T}$ is the source signal vector, and $n\left(k\right)∼\mathrm{CN}\left(0,\sigma_{n}^{2}I_{N}\right)$ is the additive white Gaussian noise vector. $A_{p}\left(\phi \right)$ corresponds to the manifold matrix of SCA [18] which is formulated as:(4)$$A_{p}\left(\phi \right)=\left[a_{p}\left(\phi_{1}\right),a_{p}\left(\phi_{2}\right),\cdots ,a_{p}\left(\phi_{D}\right)\right]\in C^{N_{p}\times D},$$
where $a_{p}\left(\phi_{d}\right)$ is the steering vector of the SCA, and the *i*-th element is given by:(5)$${\left[a_{p}\left(\phi_{d}\right)\right]}_{i}=e^{j2\pi \tilde{R}\cos\left(\phi_{d}-\beta_{i}\right)},$$
where $\tilde{R}=R/\lambda$ is the radius normalized by wavelength. Moreover, the *K* observed snapshot vectors can be packaged into a matrix as $Y_{p}=\left[y_{p}\left(1\right),y_{p}\left(2\right),\cdots ,y_{p}\left(K\right)\right]\in C^{N_{p}\times K}$. Furthermore, the sample covariance matrix of SCA in element space is calculated as:(6)$${\hat{R}}_{p}=\frac{1}{K}Y_{p}Y_{p}^{H}.$$

### 2.2. Covariance Matrix Recovery with GLS

In this subsection, we briefly review the GLS algorithm which is the underlying framework of our algorithm. The GLS algorithm proposed in [27] is a gridless extension of the sparse iterative covariance-based estimation (SPICE) [31] method. The core idea of SPICE is to perform DOA estimation based on covariance fitting. The cost function of covariance fitting is given as:(7)$$\begin{array}{cc} \; & ∥R^{-\frac{1}{2}}\left(\hat{R}-R\right){∥}_{F}^{2}\\ \; & =\mathrm{Tr}\left\{R^{-1}{\hat{R}}^{2}\right\}+\mathrm{Tr}\left\{R^{-1}\right\}-2\mathrm{Tr}\left\{\hat{R}\right\},\;K<N, \end{array}$$
and
(8)$$\begin{array}{cc} \; & ∥R^{-\frac{1}{2}}\left(\hat{R}-R\right){\hat{R}}^{-\frac{1}{2}}{∥}_{F}^{2}\\ & =\mathrm{Tr}\left\{R^{-1}\hat{R}\right\}+\mathrm{Tr}\left\{{\hat{R}}^{-1}R\right\}-2N,\;K\geq N, \end{array}$$
where $\hat{R}=\frac{1}{K}YY^{H}$ is the observed sample covariance matrix, and $Y\in C^{N\times K}$ is the matrix of snapshots. In the ULA regime where the array steering vector has a Vandermonde structure, the covariance matrix R can be re-parameterized as R=Toep{u}, which is given by:(9)$$\begin{array}{c} \mathrm{Toep}\left\{u\right\}=\left[\begin{array}{cccc} u_{1} & u_{2}^{*} & \cdots & u_{N}^{*}\\ u_{2} & u_{1} & \cdots & \;u_{N-1}^{*}\\ \vdots & \vdots & \ddots & \vdots \\ u_{N} & \;u_{N-1}\; & \cdots & u_{1} \end{array}\right]. \end{array}$$

After a series of mathematical simplifications of (Equation 7) and (Equation 8), the semidefinite problem (SDP) [32] is casted as (K<N):(10)$$\begin{array}{c} \min_{S,u}\;\mathrm{Tr}\left\{S\right\}+\mathrm{Tr}\left\{\mathrm{Toep}\left\{u\right\}\right\},\\ s.t.\;\left[\begin{array}{cc} S & \hat{R}\\ \hat{R} & \;\;\mathrm{Toep}\{u\} \end{array}\right]\geq 0, \end{array}$$
and (K≥N)
(11)$$\begin{array}{c} \min_{S,u}\;\mathrm{Tr}\left\{S\right\}+\mathrm{Tr}\left\{{\hat{R}}^{-1}\mathrm{Toep}\left\{u\right\}\right\},\\ s.t.\;\left[\begin{array}{cc} S & {\hat{R}}^{\frac{1}{2}}\\ {\hat{R}}^{\frac{1}{2}} & \;\;\mathrm{Toep}\{u\} \end{array}\right]\geq 0. \end{array}$$

Once we solve problem (Equation 10) or (Equation 11), the estimation of covariance matrix is obtained from $R_{★}=\mathrm{Toep}\left\{u_{★}\right\}$. Meanwhile, the covariance-based DOA estimation algorithm is being adopted.

In the sparse linear array (SLA) scenario, the above SDPs are extended to the following SDPs. In the case of (K<N), the SDP is given by:(12)$$\begin{array}{c} \min_{S,u}\;\mathrm{Tr}\left\{S\right\}+\mathrm{Tr}\left\{\Gamma_{\Omega }^{T}\Gamma_{\Omega }\mathrm{Toep}\left\{u\right\}\right\},\\ s.t.\;\left[\begin{array}{cc} S & {\hat{R}}_{\Omega }\\ {\hat{R}}_{\Omega } & \;\;\Gamma_{\Omega }\mathrm{Toep}\left\{u\right\}\Gamma_{\Omega }^{T} \end{array}\right]\geq 0; \end{array}$$
when K≥N, the SDP is given as:(13)$$\begin{array}{c} \min_{S,u}\;\mathrm{Tr}\left\{S\right\}+\mathrm{Tr}\left\{\Gamma_{\Omega }^{T}{\hat{R}}_{\Omega }^{-1}\Gamma_{\Omega }\mathrm{Toep}\left\{u\right\}\right\},\\ s.t.\;\left[\begin{array}{cc} S & {\hat{R}}_{\Omega }^{\frac{1}{2}}\\ {\hat{R}}_{\Omega }^{\frac{1}{2}} & \;\;\Gamma_{\Omega }\mathrm{Toep}\left\{u\right\}\Gamma_{\Omega }^{T} \end{array}\right]\geq 0, \end{array}$$
where $\Gamma_{\Omega }\in {\left\{0,1\right\}}^{N_{p}\times N}$ is the selection matrix with its entries being 1 only at the ${\left[\Gamma_{\Omega }\right]}_{n_{p},{\left[\Omega \right]}_{n_{p}}}$, and $R_{\Omega }=\frac{1}{K}\Gamma_{\Omega }YY^{H}\Gamma_{\Omega }^{T}$ is the sample covariance matrix of the SLA. Similarly, the estimated covariance matrix is obtained as $R_{★}=\mathrm{Toep}\left\{u_{★}\right\}$, which can be regarded as a completed covariance matrix of the virtual ULA. Moreover, we are able to perform DOA estimation of up to N−1 sources with the above covariance matrix. As we can see, the GLS algorithm offers a way to solve the underdetermined DOA estimation problem.

However, the Toeplitz structured covariance matrix is satisfied by ULA or virtual ULA, which is an essential precondition of the GLS algorithm. However, in the scenarios of UCA or SCA, the non-Vandermonde structured steering vector creates a big obstacle for the application of GLS algorithm. Inspired by the BT method, we extend the GLS algorithm into the SCA scenario by IBT. The IBT method is introduced in the following subsection.

### 2.3. Inverse Beamspace Transformation (IBT) of SCA

The beamspace transformation method is presented in [28], which provides a general way to reformulate the DOA estimation problem with UCA into virtual ULA. Let *M* denote the highest order mode that can be excited on a circle of normalized radius $\tilde{R}$ at a reasonable strength, which is given as:(14)$$M=\lfloor 2\pi \tilde{R}\rfloor ,$$
where ⌊·⌋ is the round-down operator. The m-th, $m\leq \left|M\right|$ phase mode is excited by the normalized beamforming vector in terms of
(15)$$b_{m}=\frac{1}{N}{\left[e^{-jm\alpha_{1}},e^{-jm\alpha_{2}},\cdots ,e^{-jm\alpha_{N}}\right]}^{T}.$$

The resulting UCA far-field beam pattern of mode m is
(16)$$\begin{array}{cc} f_{m}\left(\phi \right) & =b_{m}^{H}a\left(\phi \right)=\frac{1}{N}\sum_{n=1}^{N}e^{jm\alpha_{n}}e^{j2\pi \tilde{R}\cos\left(\phi -\alpha_{n}\right)}\\ & =j^{m}J_{m}\left(2\pi \tilde{R}\right)e^{jm\phi }+\sum_{c=1}^{\infty }\left(j^{p}J_{p}\left(2\pi \tilde{R}\right)e^{-jp\phi }+j^{q}J_{q}\left(2\pi \tilde{R}\right)e^{-jq\phi }\right), \end{array}$$
where p=cN−m and q=cN+m [29]. In order to make the first item $j^{m}J_{m}\left(2\pi \tilde{R}\right)e^{jm\phi }$ in (Equation 16) be the dominant one, the number of antennas N needs to meet the following condition:(17)N>2M.

By using the property $J_{-m}\left(2\pi \tilde{R}\right)={\left(-1\right)}^{m}J_{m}\left(2\pi \tilde{R}\right)$ of Bessel functions, and the residual terms are being omitted, the UCA beam pattern for mode m can be expressed as:(18)$$f_{m}\left(\phi \right)\approx j^{\left|m\right|}J_{\left|m\right|}\left(2\pi \tilde{R}\right)e^{jm\phi }\;\left|m\right|\leq M\;.$$

For brevity, we define the following matrix in terms of:(19)$$B=\left[b_{-M},\cdots ,b_{-1},b_{0},b_{1},\cdots ,b_{M}\right]\in C^{N\times N_{B}},$$
and
(20)$$\begin{array}{c} C_{J}=\mathrm{diag}\{j^{M}J_{M}\left(2\pi \tilde{R}\right),\cdots ,j^{1}J_{1}\left(2\pi \tilde{R}\right),j^{0}J_{0}\left(2\pi \tilde{R}\right),\\ j^{1}J_{1}\left(2\pi \tilde{R}\right),\cdots ,j^{M}J_{M}\left(2\pi \tilde{R}\right)\}\in C^{N_{B}\times N_{B}}, \end{array}$$
where $N_{B}=2M+1$ is the number of beam [20]. The Vandermonde structured array steering vector in beamspace is defined as:(21)$$a_{B}\left(\phi \right)={\left[e^{-jM\phi },\cdots ,e^{-j\phi },1,e^{j\phi },\cdots ,e^{jM\phi }\right]}^{T}.$$

By introducing $C_{J}$, (Equation 16) can be represented as: (22)$$\left[\begin{array}{c} b_{-M}^{H}\\ \vdots \\ b_{-1}^{H}\\ b_{0}^{H}\\ b_{1}^{H}\\ \vdots \\ b_{M}^{H} \end{array}\right]\cdot a\left(\phi \right)\approx C_{J}\cdot a_{B}\left(\phi \right),$$
which is
(23)$$B^{H}\cdot a\left(\phi \right)\approx C_{J}\cdot a_{B}\left(\phi \right).$$

Obviously we have:(24)$$a\left(\phi \right)\approx {\left(B^{H}\right)}^{†}C_{J}\cdot a_{B}\left(\phi \right)=T_{B}\cdot a_{B}\left(\phi \right).$$

To sum up, the relation between a(ϕ) and $a_{v}\left(\phi \right)$ is given as:(25)$$a\left(\phi \right)\approx T_{B}a_{B}\left(\phi \right),$$
where $T_{B}$ is defined as
(26)$$T_{B}={\left(B^{H}\right)}^{†}C_{J}.$$

Apparently, $T_{B}$ transforms the steering vectors in beamspace into element space, which is exactly the reverse of the original beamspace transformation. Therefore, the above process is named **I**nverse **B**eamspace **T**ransformation (IBT).

As we can see, (Equation 25) offers great convenience for handling the non-Vandermonde structured steering vector of UCA which is the basic framework of our algorithm.

## 3. Proposed Algorithm GSCA

The proposed algorithm named GSCA (**G**ridless DOA Estimation in **S**parse **C**ircular **A**rray) is summarized in Algorithm 1. In order to visualize the principle of the algorithm, we draw the main steps of the algorithm in Figure 2.
**Algorithm 1:** *Proposed Algorithm*: **GSCA****Input:** $N_{p}$, *N* , $N_{B}$ , *D* , *R* , $Y_{p}$ , $\Gamma_{\Omega }$**Output:** Estimated DOAs $\left\{{\hat{\phi }}_{1},{\hat{\phi }}_{2},\cdots ,{\hat{\phi }}_{D}\right\}$**Step 1:** Calculate ${\hat{R}}_{p}$ via (Equation 6),**Step 2:** Calculate $T_{B}$ via (Equation 25),**Step 3: If** $K<N_{B}$, perform (Equation 31) ;**Else** $K\geq N_{B}$, perform (Equation 32),**Step 4:** Formulate ${\tilde{R}}_{v}$,and calculate its EVD via (Equation 34),**Step 5:** Perform Root-MUSIC based on (Equation 37)–(Equation 40),**Step 6:** Return DOAs via (Equation 41).

Next, the GSCA algorithm is introduced in detail. The GSCA algorithm is mainly based on the GLS algorithm combined with the aforementioned IBT. Notably, the elements of SCA are selected from a UCA, with utilizing the predefined selection matrix $\Gamma_{\Omega }$, the relation between the physical snapshot of SCA and the complete snapshot of UCA is established as:(27)$$y_{p}=\Gamma_{\Omega }y.$$

Thus, the covariance matrix of SCA is written as:(28)$$\begin{array}{cc} R_{p} & =E\{y_{p}y_{p}^{H}\}\\ \; & =\Gamma_{\Omega }E\left\{yy^{H}\right\}\Gamma_{\Omega }^{T}\\ \; & =\Gamma_{\Omega }R\Gamma_{\Omega }^{T}, \end{array}$$
where
(29)$$\begin{array}{cc} R=E\{yy^{H}\} & =E\{\left(As+n\right){\left(As+n\right)}^{H}\}\\ & =AE\left\{ss^{H}\right\}A^{H}+E\left\{nn^{H}\right\}\\ & =AR_{s}A^{H}+R_{n}\\ & =R_{x}+R_{n}, \end{array}$$
with $E\left\{sn^{H}\right\}=E\left\{s^{H}n\right\}=0$ being used; meanwhile, $R_{s}$ and $R_{n}$ are the signal covariance matrix and noise covariance matrix, respectively. By taking advantage of the IBT, matrix $R_{x}$ can be reparameterized as:(30)$$\begin{array}{cc} R_{x} & =AR_{s}A^{H}\\ & \approx T_{B}A_{v}R_{s}A_{v}^{H}T_{B}^{H}\\ & =T_{B}\mathrm{Toep}\left\{u\right\}T_{B}^{H}. \end{array}$$

Similarly, with an application of the GLS algorithm, the SDPs arising in the SCA scenario are shown below. In the case of ($K<N_{B}$), the SDP is given as:(31)$$\begin{array}{c} \min_{S,u}\;\mathrm{Tr}\left\{S\right\}+\mathrm{Tr}\left\{\Gamma_{\Omega }^{T}\Gamma_{\Omega }T_{B}\mathrm{Toep}\left\{u\right\}T_{B}^{H}\right\},\\ s.t.\;\left[\begin{array}{cc} S & {\hat{R}}_{p}\\ {\hat{R}}_{p} & \;\;\Gamma_{\Omega }T_{B}\mathrm{Toep}\left\{u\right\}T_{B}^{H}\Gamma_{\Omega }^{T} \end{array}\right]\geq 0. \end{array}$$

When $K\geq N_{B}$, the SDP is given as:(32)$$\begin{array}{c} \min_{S,u}\;\mathrm{Tr}\left\{S\right\}+\mathrm{Tr}\left\{\Gamma_{\Omega }^{T}{\hat{R}}_{p}^{-1}\Gamma_{\Omega }T_{B}\mathrm{Toep}\left\{u\right\}T_{B}^{H}\right\},\\ s.t.\;\left[\begin{array}{cc} S & {\hat{R}}_{p}^{\frac{1}{2}}\\ {\hat{R}}_{p}^{\frac{1}{2}} & \;\;\Gamma_{\Omega }T_{B}\mathrm{Toep}\left\{u\right\}T_{B}^{H}\Gamma_{\Omega }^{T} \end{array}\right]\geq 0. \end{array}$$

Once problem (Equation 31) or (Equation 32) is solved, the completed covariance matrix of the UCA in element space is obtained as $R_{★}=T_{B}\mathrm{Toep}\left\{u_{★}\right\}T_{B}^{H}$. The middle part of $R_{★}$ is a Toeplitz structured covariance which can be regarded as a beamspace transformed covariance matrix of the UCA. Next, we focus on the middle part which is marked as ${\tilde{R}}_{B}=\mathrm{Toep}\left\{u_{★}\right\}$. Apparently, a classical Root-MUSIC algorithm [33] can be performed thanks to the Toeplitz structure of ${\tilde{R}}_{B}$.

The eigenvalue decomposition (EVD) of ${\tilde{R}}_{B}$ is given as:(33)$${\tilde{R}}_{B}=U_{s}\Lambda_{s}U_{s}^{H}+U_{n}\Lambda_{n}U_{n}^{H},$$
where the signal subspace $U_{s}$, noise subspace, $U_{n}$ and corresponding eigenvalues $\Lambda_{s}$, $\Lambda_{n}$ have the following forms:(34)$$\begin{array}{cc} U_{s} & =\left[u_{1},\cdots ,u_{D}\right]\;\;\Lambda_{s}=\mathrm{diag}\left\{\lambda_{1},\cdots ,\lambda_{D}\right\}\\ U_{n} & =\left[u_{D+1},\cdots ,u_{N_{B}}\right]\;\Lambda_{n}=\mathrm{diag}\left\{\lambda_{D+1},\cdots ,\lambda_{N_{B}}\right\}. \end{array}$$

As we all know, the noise subspace $U_{n}$ is orthogonal to the signal subspace $U_{s}$, and $U_{s}$ spans the same subspace as the steering matrix which is written as
(35)$$A_{B}=\left[a_{B}\left(\phi_{1}\right),a_{B}\left(\phi_{2}\right),\cdots ,a_{B}\left(\phi_{D}\right)\right].$$

It is obvious to formulate the following equation:(36)$$U_{n}⊥A_{B}\Leftrightarrow U_{n}A_{B}=0.$$

The null spectrum is formed as:(37)$$\begin{array}{cc} f(z) & =∥U_{n}a_{B}{\left(\phi \right)∥}_{2}^{2}=a_{B}{\left(\phi \right)}^{H}U_{n}U_{n}^{H}a_{B}\left(\phi \right),\\ \; & =p{\left(\frac{1}{z}\right)}^{T}U_{n}U_{n}^{H}p\left(z\right), \end{array}$$
for $z=e^{j\phi }$, and p(z) is defined as:(38)$$p\left(z\right)={\left[1,z,\cdots ,z^{N_{B}-1}\right]}^{T}.$$

Moreover, the null spectrum is also able to reformulate into a polynomial as follows:(39)$$G\left(z\right)=h_{-(N_{B}-1)}z^{-(N_{B}-1)}+\cdots +h_{\left(N_{B}-1\right)}z^{\left(N_{B}-1\right)},$$
where $h_{i}$ is calculated by:(40)$$\begin{array}{cc} H & =U_{n}U_{n}^{H}\\ h_{i} & =\sum_{n_{1},n_{2}}^{N_{B}}{\left[H\right]}_{n_{1},n_{2}},\;n_{1}-n_{2}=i. \end{array}$$

The *D* roots inside the unit circle with the largest magnitude are chosen.Then the DOAs are obtained by:(41)$${\hat{\phi }}_{d}=∠z_{d}.$$

## 4. Simulation Results

In this section, computer simulations are carried out to demonstrate the performance of the proposed DOA estimation algorithms. The root-mean-square error (RMSE) is adopted, which is defined as:(42)$$\mathrm{RMSE}=\sqrt{\frac{1}{PD}\sum_{p=1}^{P}\sum_{d=1}^{D}{\left({\hat{\phi }}_{d}\left(p\right)-\phi_{d}\left(p\right)\right)}^{2}},$$
where P=200 is the number of Monte Carlo trials.

Additionally, the spatial spectrum is depicted in the polar coordinate to visualize the performance of DOA estimation. To obtain the spatial spectrum, we replace step 5 of the GSCA algorithm in Algorithm 1 (Root-MUSIC [33]) with a spatial spectrum search (MUSIC [5]). The spatial search step is $0.1^{\circ }$, and the search range is $\left[0^{\circ },360^{\circ }\right)$.

### 4.1. Selection of *N*, $N_{p}$ and Ω

In this subsection, the simulation results are presented to illustrate the selection of *N*, $N_{p}$, and Ω. We explored the effect of various $N_{p}$ on the performance of DOA estimation under a selected *N*. The SNR and *K* are chosen as 15 dB and 1024, and the number of Monte Carlo trials *P* is 200. The number of sources *D* and the number of physical elements $N_{p}$ are equal in order to satisfy the underdetermined scenario. Obviously, multiple label sets Ω will be generated under each pair of $\left(N,N_{p}\right)$. In order to exclude the influence of the particularity of Ω on the results, label set Ω is randomly generated in each trial. We set a threshold of RMSE (Equation 43) to evaluate the simulation results. The simulation results are shown in parts a–c of Table 2, respectively. (The minimum $N_{p}$ that succeeds under each *N* is bolded.)
(43)$$\mathrm{RMSE}\left\{\begin{array}{cc} <8^{\circ } & \;Success\\ \geq 8^{\circ } & \;Failed \end{array}\right.$$

From the above results, we can roughly draw the following empirical conclusions to select $N_{p}$, which is given by:(44)$$N_{p}\geq \left\lceil \frac{2}{3}N\right\rceil ,$$
where ⌈·⌉ is the round-up operator. To sum up, we choose the minimum $N_{p}$ that succeeds when N=7 or 9.

Apparently, when sensor failure occurs, the locations of the faulty sensors are random. Considering the circular symmetry, there are three different array configurations when N=7 and $N_{p}=5$; and the number of different array configurations is seven when N=9 and $N_{p}=6$. We have plotted these arrays in Figure 3 and Figure 4, respectively.

We perform the following simulations to evaluate the performance of the GSCA algorithm under different array configurations when the SNR and the number of snapshots *K* are fixed at a moderate value (SNR = 15 dB, *K* = 1024). The number of sources *D* is an integer variable selected from $\left[N_{p},N\right)$ to satisfy the underdetermined scenario. The simulation results are shown in Figure 5a and Figure 5b, respectively. Obviously, when the simulation parameters {SNR$,K,D,N,N_{p}$} are the same, the RMSEs of different array configurations are almost at the same level, which means the robustness of our method to various array configurations with the same *N* and $N_{p}$. Therefore, in the following simulations, we selected one type of array from each group to study the effects of SNR and *K*.

### 4.2. Effects of SNR and *K*: N=7, $N_{p}=5$

In this simulation, we study the performance of DOA estimation versus SNR and *K*. The number of sensors in UCA: *N* is set to 7, and the number of physical sensors $N_{p}$ is set to 5 with label Ω={1,2,3,5,7} (Type 2 in Figure 3). A schematic of this SCA is shown in Figure 3b. The normalized radius $\tilde{R}$ is set to 0.65, in this case, $N_{B}$ is 7 based on (Equation 14). Notably, the source number *D* is set to 5 and 6 in order to perform underdetermined DOA estimation, and DOAs are set to be equidistantly distributed in $\left[0^{\circ },360^{\circ }\right)$ [34].

The RMSE versus SNR and number of snapshots *K* are plotted in Figure 6, respectively. As shown in Figure 6a, the RMSE is gradually dropping as the SNR increases; yet the RMSE is slightly dropping as *K* increases. In addition, the RMSE increases as the number of sources *D* increases when the SNR and *K* are fixed (Figure 6b).

Furthermore, Figure 7a,b depicts the normalized spatial spectrums under D=5 and D=6, respectively. The SNR is set to be 10 dB, and the number of snapshots *K* is 1024. Apparently, as the number of sources *D* increases, the number of outlier peaks increases when SNR and the number of snapshots *K* are fixed.

### 4.3. Effects of SNR and *K*: N=9, $N_{p}=6$

In this simulation, the number of sensors in UCA, *N*, is set to 9, and the number of physical sensors $N_{p}$ is set to 6. The index vector of the physical sensor is Ω={1,2,3,5,7,9} (Type 5 in Figure 3). The normalized radius $\tilde{R}$ is set to 0.7, and $N_{B}$ is calculated as 9 with (Equation 14). The array structure is shown in Figure 4e. In contrast with the simulation of N=7, $N_{p}=5$, the RMSE curves and the normalized spatial spectrum are drawn in Figure 8 and Figure 9, respectively. Moreover, the RMSE curve of D=6 in Figure 8a converges to $1.2^{\circ }$ while the RMSE curve of D=6 in Figure 6a converges to $2.8^{\circ }$. Furthermore, comparing Figure 7b and Figure 9a, it is evident that the number of outlier peaks decreases when *D* is fixed as 6. Those benefits come with the enlarged array aperture $\tilde{R}$ and the increased number of physical sensors $N_{p}$.

### 4.4. Complexity Analysis

The major computational complexity of the proposed GSCA algorithm corresponds to the step (Equation 31) or (Equation 32). A well-known off-the-shelf SDP solver SDPT3 [35] is employed to solve our algorithm. The SDPT3 is an interior-point-based method that has the computational complexity of $O(n_{1}^{2}n_{2}^{2.5})$ [36], where $n_{1}$ denotes the number of variables and $n_{2}\times n_{2}$ is the dimension of the positive semi-definite matrix in the SDP. In our cases, $n_{1}=N_{B}+N_{p}^{2}$ and $n_{2}=2N_{p}$, such that the computational complexity of solving SDP is $O({\left(N_{B}+N_{p}^{2}\right)}^{2}{\left(2N_{p}\right)}^{2.5})$. In addition, the EVD (Equation 34) step also contributes a large part of the computational complexity, which is $O(N_{B}^{3})$. The complexity of polynomial rooting steps (Equation 37)–(Equation 41) is $O(DN_{B})$. Thus the major computational complexity of the proposed GSCA algorithm is $O({\left(N_{B}+N_{p}^{2}\right)}^{2}{\left(2N_{p}\right)}^{2.5}+N_{B}^{3}+DN_{B})$. We evaluate the algorithm under CPU I7-10510U at 2.30 GHz and 12 GB RAM. The average CPU running time of 200 Monte-Carlo trials is given in parts a and b in Table 3, respectively.

## 5. Conclusions

In this paper, we have proposed the GSCA algorithm to perform underdetermined DOA estimation in SCA. The GSCA algorithm takes advantage of the inverse beamspace transformation (IBT), together with the GLS algorithm; in this way, the covariance matrix of SCA is completed to a Toeplitz matrix in beamspace; meanwhile, the Root-MUSIC is adopted and DOAs are obtained. We have performed computer simulations, and results demonstrate that the proposed algorithm is able to produce reasonable results of underdetermined DOA estimation in SCA. Furthermore, the GSCA algorithm still works well in various array configurations, which means the tricky DOA estimation problem in UCA with random sensor failures can be handled. In the future, we will work on improving the performance of the algorithm and strive to extend our algorithm to the two-dimensional DOA estimation problem.

## Figures and Tables

**Figure 1 sensors-22-02864-f001:**
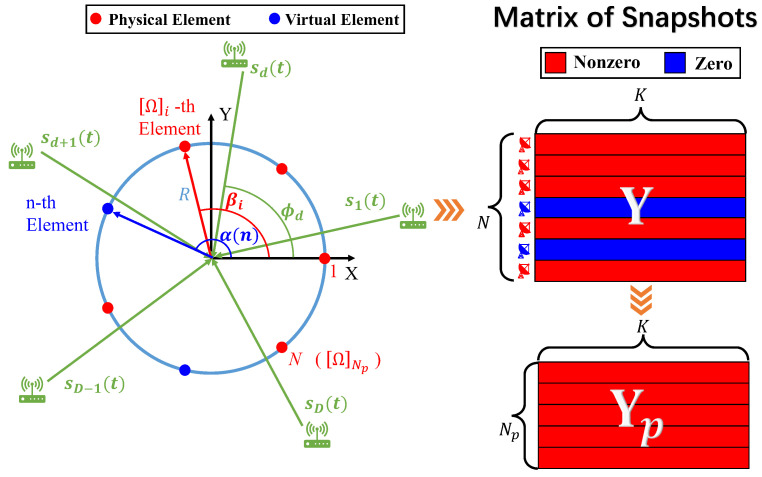
System Model.

**Figure 2 sensors-22-02864-f002:**
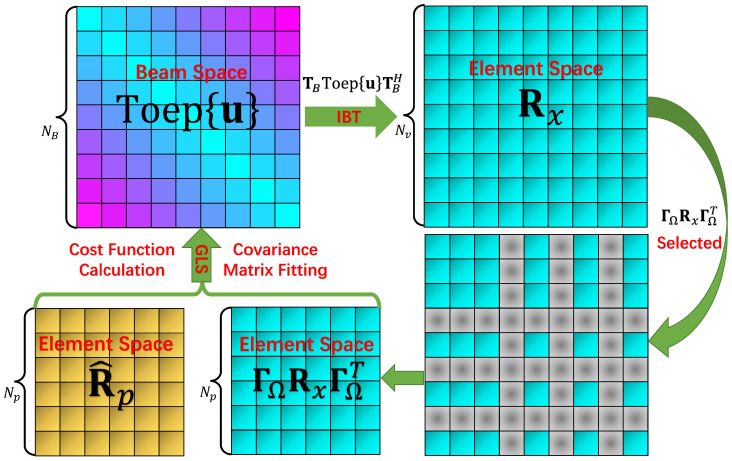
Schematic of Proposed GSCA Algorithm.

**Figure 3 sensors-22-02864-f003:**
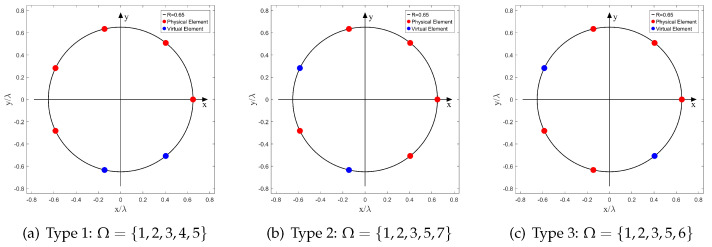
Different Array Configurations, (*N* = 7, $N_{p}$ = 5).

**Figure 4 sensors-22-02864-f004:**
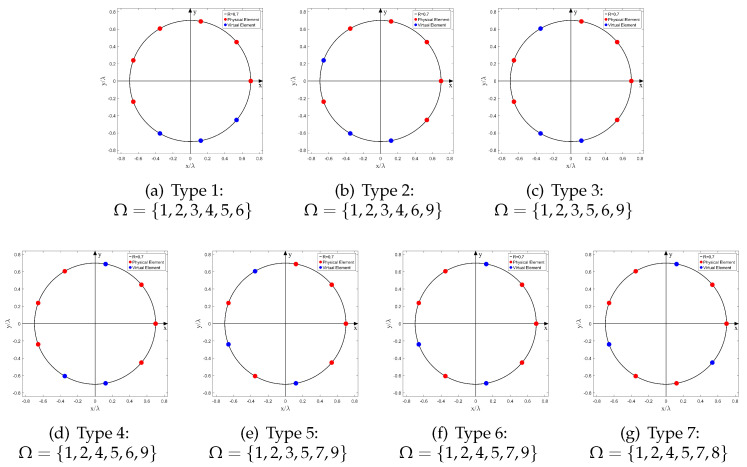
Different Array Configurations, (*N* = 9, $N_{p}$ = 6).

**Figure 5 sensors-22-02864-f005:**
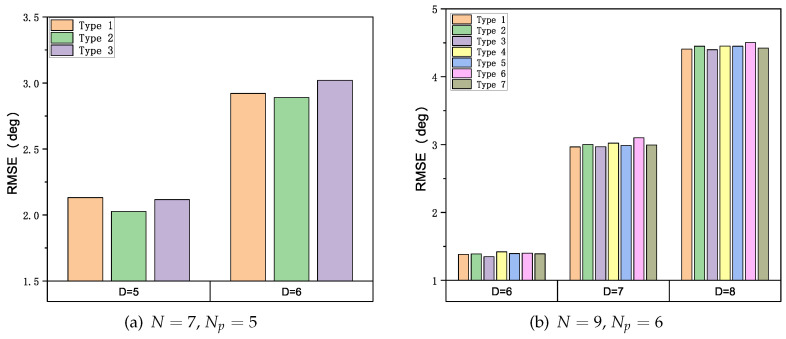
RMSE under Different Array Configurations; SNR = 15 dB, *K* = 1024.

**Figure 6 sensors-22-02864-f006:**
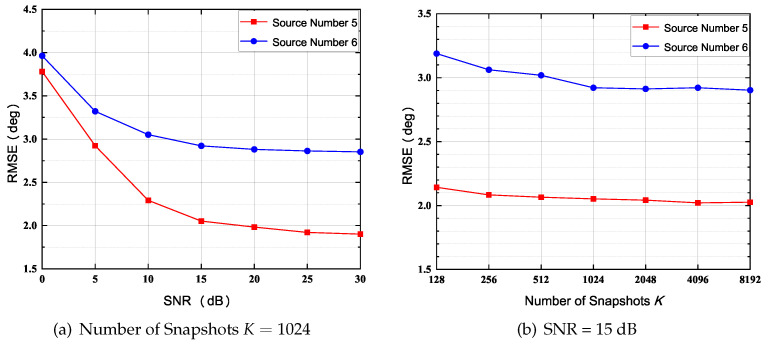
RMSE of 5 and 6 Sources, (*N* = 7, $N_{p}$ = 5).

**Figure 7 sensors-22-02864-f007:**
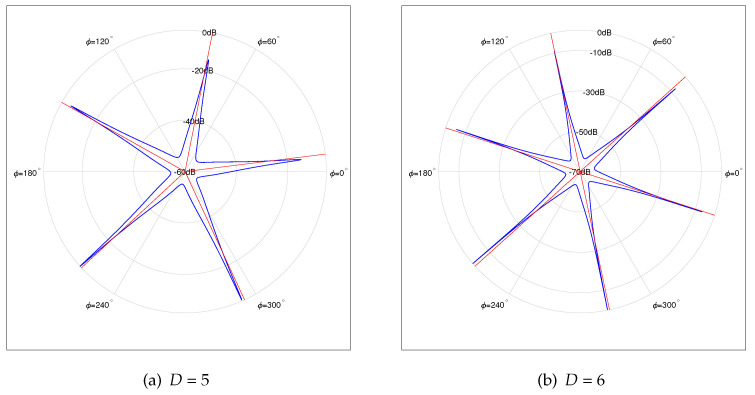
Normalized Spatial Spectrum of 5 and 6 Sources, *N* = 7, $N_{p}$ = 5; (SNR=10 dB, *K* = 1024).

**Figure 8 sensors-22-02864-f008:**
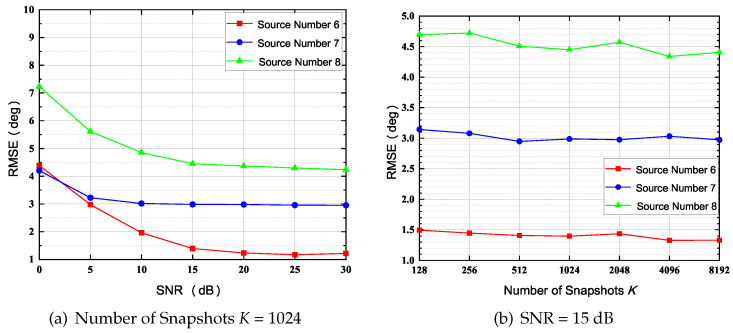
RMSE of 6, 7 and 8 Sources, (*N* = 9, $N_{p}$ = 6).

**Figure 9 sensors-22-02864-f009:**
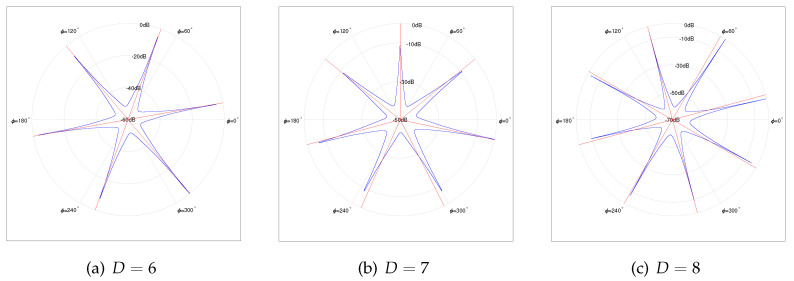
Normalized Spatial Spectrum of 6, 7 and 8 Sources, *N* = 9, $N_{p}$ = 6; (SNR = 10 dB, *K* = 1024).

**Table 1 sensors-22-02864-t001:** Differences and Connections.

Reference	Array Geometry Scenario	Core Method	Others
Yin et al. [19]	ULA Determined	Sparse Representation of Array Covariance Vectors	Grid
Zhao et al. [20]	UCA Determined	BT; Sparse Representation of Array Covariance Vectors	Grid
Jiang et al. [18]	Nested SCA Underdetermined	Sparse Representation of Array Covariance Vectors	Grid
Yadav et al. [30]	Rotate SCA Underdetermined	Sparse Representation of Array Covariance Vectors	Grid
Our Work	SCA Underdetermined	IBT; Covariance Matrix Recovery with GlS	Gridless

**Table 2 sensors-22-02864-t002:** Simulation Results of Various $\left(N,N_{p}\right)$.

(a) *N* = 7, SNR = 15 dB, *K* = 1024						
	$N_{p}\left(D\right)$	3	4	**5**	6	
	Results	Failed	Failed	**Success**	Success	
(b) *N* = 9, SNR = 15 dB, *K* = 1024						
	$N_{p}\left(D\right)$	4	5	**6**	7	8
	Results	Failed	Failed	**Success**	Success	Success
(c) *N* = 11, SNR = 15 dB, *K* = 1024						
$N_{p}\left(D\right)$	5	6	**7**	8	9	10
Results	Failed	Failed	**Success**	Success	Success	Success

**Table 3 sensors-22-02864-t003:** CPU Running Time (SNR = 15 dB, *K* = 1024).

(a) $N=7,N_{p}=5$, Various Ω			(b) $N=9,N_{p}=6$, Various Ω			
*D*	5	6	*D*	6	7	8
Time (s)	2.902	2.918	Time (s)	3.102	3.084	3.195

## Data Availability

Not applicable.

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
