# Peer review of "Gridless Underdetermined Direction of Arrival Estimation in Sparse Circular Array Using Inverse Beamspace Transformation"

_sensors, 2022, doi:10.3390/s22082864_

Round 1

Reviewer 1 Report

Dear Authors,

   in the way you have written it, your paper is more a technical note than an article. You created, modelled, and tested an algorithm. I don't think the relevance goes beyond a technical note. Furthermore, you did not take into account some already published articles that could have been useful to introduce the problem (for example: 1. https://www.mdpi.com/1424-8220/19/20/4427/htm 2. https://ieeexplore.ieee.org/stamp/stamp.jsp?tp=&arnumber=5783354 3. https://asp-eurasipjournals.springeropen.com/articles/10.1186/s13634-021-00770-2 4. https://www.mdpi.com/1424-8220/20/8/2222/htm).

Moreover, a pragmatic approach is missing in both the introduction and the conclusions, indicating at least in theory the practical use, in the real world, of such an algorithm. I am not saying that a technical-practical proof was needed, but at least a hint of possible future practical applications.

Finally, the paper needs to be reviewed by a native English speaker specialising in the field.

For these reasons I suggest a major revision.

Reviewer 2 Report

- The text requires proof reading as there are some typos, some articles are missing, some verbs are not in the correct sense.

- The state of the art must be expanded and improved. Also, a table for this task should be added. For each manuscript, its contributions and conclusions must be exposed. The motivation of the paper is unclear, please add examples.

- In section 2, references must be inserted to validate equations. Furthermore, a figure of the system model must be presented and explained.

- Two scenarios are exposed, they must be justified.

- Comparison results against the benchmarking techniques in terms of performance could be added. Also, the complexity study with its discussion must be exposed.

- Only, numerical results are presented. Experimental observations must be inserted to validate the research work. Number of Snapshots and SNR are fixed in the presentations of the results; in order to generalization observations, these parameters must be variables

- Future research direction must be added.

- Some references could be more recent, some of them are more than 5 years old.

- Abbreviations must be inserted according to the template of the journal.

Round 2

Reviewer 1 Report

Dear Authors,

I am OK with your amendaments. I can accept your paper in the current final version.

Reviewer 2 Report

Accept in present form